# Evaluating Iso-Mukaadial Acetate and Ursolic Acid Acetate as *Plasmodium falciparum* Hypoxanthine-Guanine-Xanthine Phosphoribosyltransferase Inhibitors

**DOI:** 10.3390/biom9120861

**Published:** 2019-12-11

**Authors:** Francis Opoku, Penny P. Govender, Ofentse J. Pooe, Mthokozisi B.C. Simelane

**Affiliations:** 1Department of Chemical Sciences, University of Johannesburg, P.O. Box 17011, Doornfontein Campus, Johannesburg 2028, South Africa; pennyg@uj.ac.za; 2Discipline of Biochemistry, School of Life Science, University of KwaZulu-Natal, Westville 4000, South Africa; PooeO@ukzn.ac.za; 3Department of Biochemistry, Faculty of Science, University of Johannesburg, Johannesburg 2006, South Africa

**Keywords:** iso-mukaadial acetate, ursolic acid acetate, *Pf*HGXPT, *Plasmodium falciparum*

## Abstract

To date, *Plasmodium falciparum* is one of the most lethal strains of the malaria parasite. *P. falciparum* lacks the required enzymes to create its own purines via the de novo pathway, thereby making *Plasmodium falciparum* hypoxanthine-guanine-xanthine phosphoribosyltransferase (*Pf*HGXPT) a crucial enzyme in the malaria life cycle. Recently, studies have described iso-mukaadial acetate and ursolic acid acetate as promising antimalarials. However, the mode of action is still unknown, thus, the current study sought to investigate the selective inhibitory and binding actions of iso-mukaadial acetate and ursolic acid acetate against recombinant *Pf*HGXPT using in-silico and experimental approaches. Recombinant *Pf*HGXPT protein was expressed using *E. coli* BL21 cells and homogeneously purified by affinity chromatography. Experimentally, iso-mukaadial acetate and ursolic acid acetate, respectively, demonstrated direct inhibitory activity towards *Pf*HGXPT in a dose-dependent manner. The binding affinity of iso-mukaadial acetate and ursolic acid acetate on the *Pf*HGXPT dissociation constant (KD), where it was found that 0.0833 µM and 2.8396 µM, respectively, are indicative of strong binding. The mode of action for the observed antimalarial activity was further established by a molecular docking study. The molecular docking and dynamics simulations show specific interactions and high affinity within the binding pocket of *Plasmodium falciparum* and human hypoxanthine-guanine phosphoribosyl transferases. The predicted in silico absorption, distribution, metabolism and excretion/toxicity (ADME/T) properties predicted that the iso-mukaadial acetate ligand may follow the criteria for orally active drugs. The theoretical calculation derived from ADME, molecular docking and dynamics provide in-depth information into the structural basis, specific bonding and non-bonding interactions governing the inhibition of malarial. Taken together, these findings provide a basis for the recommendation of iso-mukaadial acetate and ursolic acid acetate as high-affinity ligands and drug candidates against *Pf*HGXPT.

## 1. Introduction

Even though great strides have been taken to control malaria over the past decade, it is still regarded as a major public health problem across the globe [1]. According to the World Health Organization, malaria continues to be endemic throughout developing countries, with over 200 million cases and approximately 450,000 deaths in 2017 [2]. The continued development of antimalarial drug resistance has necessitated the need for the development of novel drugs. Medicinal plants are known to have secondary metabolites that possess different biological activities. A number of compounds isolated from plants have not been studied fully, particularly their mode of action. It is estimated that 25% of drugs are derived from plants [3], therefore, investigating these compounds could possibly lead to the discovery of more potent drugs that could be used against malaria parasites. Using in vitro and in vivo experiments, our research group reported new plant-derived antimalarial compounds, including iso-mukaadial acetate and ursolic acid acetate [4,5]. However, the exact antimalarial mechanism of action for these compounds is still unknown. In this study, we investigated *Plasmodium falciparum* hypoxanthine-guanine-xanthine phosphoribosyltransferase (*Pf*HGXPRT), an essential malaria protein, as a potential target for iso-mukaadial acetate and ursolic acid acetate. The *Plasmodium falciparum* parasite lacks the necessary de novo pathway enzymes to create its own purines and hence is dependent upon the salvage pathway [6,7,8]. Thus this difference can be exploited to design novel drugs to inhibit *Pf*HGXPRT [9,10,11]. In this current report, we describe for the first time the direct inhibitory role of iso-mukaadial acetate (IMA) and ursolic acid acetate (UAA) on *Pf*HGXPRT using computational and various wet-lab approaches.

Molecular docking is a theoretical technique which is used to predict the protein–ligand interaction, as well as to characterise the behaviour of the ligands in the binding pocket of the target receptor [12]. Up to now, molecular docking has gained much attention in identifying the possible binding modes and affinities occurring between a ligand and a target receptor or protein. Herein, we offer in-depth insights into the binding modes and protein–ligand complexation of iso-mukaadial acetate and ursolic acid acetate with *Plasmodium falciparum* and human hypoxanthine-guanine phosphoribosyl transferases receptor using molecular docking, Prime/Molecular Mechanics-Generalized Born Surface Area (Prime/MM-GBSA) binding free energy calculation and molecular dynamics (MD) simulation.

## 2. Materials and Methods

Two recently described antimalarial compounds were used in this study: iso-mukaadial acetate, with a molecular weight of 308.36 g/mol, and UAA, with a molecular weight of 498.748 g/mol [4,13]. The genetic material encoding *Pf*HGXPRT was synthesized by Genscript Corporation (Nanjing, China) and all other chemicals used in this study were purchased from Sigma-Aldrich Co. Ltd. (St. Louis, MO, USA), unless specifically stated.

### 2.1. Recombinant Expression and Purification of PfHGXPRT

A codon harmonized form of the gene encoding the *Pf*HGXPRT (PlasmoDB accession number: PF3D7_1012400.1) gene was synthesized by Genscript Corporation (Nanjing, China). The gene was subsequently cloned into pET28-a (Qiagen, Hilden, Germany) in frame with the N-termini 6X Histidine (His6) encoding sequence to facilitate protein purification. The recombinant *Pf*HGXPRT gene was overexpressed in *Escherichia coli* BL21 (DE3) cells using the bacteriophage T5 RNA polymerase and promoter system. The cells containing the recombinant gene were grown in 1000 mL LB broth at 37 °C, containing 50 μg/mL kanamycin. At A600 = 0.6, the expression was induced using 1 mM isopropyl-β-D-1-thiogalactopyranoside (IPTG). After being grown for more than 24 h, the cells were harvested by centrifugation and the pellet was suspended in LEW buffer (50 mM NaH_2_PO_4_, 300 mM NaCl–HC, pH 7.4), 1 mM phenylmethylsulfonyl fluoride (PMSF; Roche, Mannheim, Germany) and 1 mg/mL lysozyme and incubated at 23 °C for 1 h. The crude cell extract was further lysed by low-speed homogenization for 5 min. The cellular debris was then removed by centrifugation at 8000× g for 20 min at 4 °C, and the crude lysate extract was then used for *Pf*HGXPRT purification. The supernatant was loaded onto nickel-charged protino nitrilotriacetic acid (Ni-NTA; Macherey-Nagel, GmbH, Germany) for 1 h on ice to enable efficient His-tagged protein affinity binding. Following the manufacturer’s protocols, the Ni-NTA was then washed using LEW. The bound protein was eluted using a LEW buffer containing 250 mM imidazole. The eluted protein fractions were then visualized by 12.5% sodium dodecyl sulphate–polyacrylamide gel electrophoresis (SDS-PAGE). Protein concentrations were estimated by Bradford assay (Sigma-Aldrich, USA) following the manufacturer’s instructions, using bovine serum albumin as protein standard. The production of the His6 protein was confirmed using α-His horse-raddish-peroxidase (HRP) conjugated antibodies (Sigma-Aldrich, USA) at 1:2000 dilution; colourimetric–based protein detection on Nitrocellulose membranes (Merck, Germany) was done using the 3,3′,5,5′-Tetramethylbenzidine (TMB) Liquid Substrate System for Membranes (Sigma-Aldrich).

### 2.2. UV-Vis Spectrometer Analysis

Several techniques have been developed in an effort to try and better understand protein–protein and protein–ligand interaction, as the basis of protein interaction opens the doorway to understanding vital reactions in organisms [14]. The study of proteins binding to ligands requires that we study and understand how proteins interact with drugs in terms of their conformation, binding affinities, kinetics and thermodynamics, in an effort to develop specific drugs that target proteins that have a vital role in the diseases and illness [7,14]. UV-Vis spectrophotometry can be used to monitor protein secondary structure changes through three aromatic amino acids: Phe, Tyr and Trp. The conformational changes induced by the inhibitors were monitored by UV–Vis–NIR spectrophotometer (V-730 BIO; Jasco, Tokyo, Japan). Vials containing 20 µg/mL of *Pf*HGXPRT were, respectively, exposed to 0, 5 and 10 mM concentrations of iso-mukaadial acetate and ursolic acid acetate for 15 min. *Pf*HGXPRT was similarly exposed to the commercial standard inhibitor, Triethyl-2-phosphonobutyrate (Tri). The *Pf*HGXPRT spectra were collected at room temperature before and after the addition of all inhibitors.

### 2.3. Binding Affinity Analysis Using MicroScale Thermophoresis 

Microscale thermophoresis (MST) has previously been reported in the investigation of various intramolecular interactions between biological macromolecules in solution [14,15]. Herein, MST was employed to investigate the interaction between *Pf*HGXPRT and the compounds. Protein and ligand labelling was performed following the manufacturer’s protocols with minor adjustments. Briefly, a 200 nM solution of *Pf*HGXPRT was labelled using the His-Tag Labelling Kit Red-tris-NTA dye at room temperature for 30 min (NanoTemper Technologies, GmbH, Germany). The labelled protein was then diluted to 10 μM using the MST buffer with 0.05% Tween 20 (Nano Temper Technologies, GmbH, Germany). Using the same buffer, the experimental ligands were prepared in a series of 16 dilutions, 5−0.000153 μM for UAA and 1−3.05 × 10^5^ μM MA. Each of the ligands was then co-incubated for 10 min in the presence of the labelled *Pf*HGXPRT (1:1 ratio). The samples were then loaded onto the Monolith NT.115 instrument using standard treated capillaries (MO-L014) (Nano Temper technologies GmbH, Germany). The Monolith NT.115 (Nano Temper Technologies, GmbH, Germany) was used at a room temperature of 25 °C, excitation power 20 and 40% MST power settings resulting in the production of MST Traces. The binding assays were conducted in triplicates and analysed using the MO-Affinity Analysis software version 2.1.3 (Nano Temper Technologies, GmbH, Germany).

### 2.4. Computational Details

Molecular docking and molecular dynamics studies were conducted to investigate the computational binding affinity between *Pf*HGXPT and the two compounds. Molecular docking was carried out using Schrödinger Release 2019-2 to compute the binding affinity of the iso-mukaadial acetate and ursolic acid acetate complexes with *Plasmodium falciparum* and human hypoxanthine-guanine phosphoribosyl transferases protein (PDB-ID: 2VFA) [16]. The 2VFA structure was obtained from the Protein Data Bank (https://www.rcsb.org/structure) [17]. For comparison, molecular docking and molecular dynamics studies on human hypoxanthine-guanine phosphoribosyltransferase (HGPRT) was considered. The crystal structure of the free human hypoxanthine-guanine phosphoribosyltransferase (HGPRT) receptor with PDB-ID: 1Z7G [18]) was obtained from the Research Collaboratory for Structural Bioinformatics (RCSB) Protein Data Bank (https://www.rcsb.org/structure) [17]. Solution studies on 2VFA have shown that it occurs as a mixture of monomer and dimer, and undergoes tetramerisation on the addition of phosphoribosyl pyrophosphate [16]. On the other hand, both *Pf*HGXPT and human HGPRT are tetramers in all liganded states. In addition, 2VFA also shows additional specificity for xanthine, similar to Domain Switch 1 (DS1) and PfHGPRT. DS1 was the first studied chimeric enzyme, which contained the 49 terminal amino acids residues from *Pf*HGXPT and the rest from human HGPRT [19]. The crystal structure of a chimera of *Pf*HGXPT, which consists of the core of the hood region from the Pf enzyme and the protein from the human enzyme, has been determined as a complex with the product guanosine monophosphate [16]. The chimera can utilise xanthine, guanine and hypoxanthine as substrates, which is similar to the *Plasmodium falciparum* enzyme. The chimera occurs as a monomer–dimer mixture in solution, but shifts to a tetramer with the addition of phosphoribosyl pyrophosphate [16].

### 2.5. Preparation of the Targeted Protein

The receptor was prepared for docking using the Schrödinger Release 2019-2 Protein Preparation Wizard [20]. After ensuring chemical accuracy, Epik [21] was used to add hydrogen atoms and tautomeric states at a pH 7.0 ± 2.0, as well as to neutralise the side chains that were not close to the active sites or actively involved in the formation of a salt bridge. This was followed by filling missing side loops and chains using Prime module [22]. Hydrogen atoms that were added to the structure and water molecules which were 3 Å away from the ligand were removed using PROPKA [23] at pH 7.0. Following the assignment of the appropriate charges, bond orders and atom types, restrained optimisation was carried out using the OPLS3e force field [24] to achieve a maximum root-mean-square deviation (RMSD) of 0.30 Å.

### 2.6. Ligand Preparation 

The structures of iso-mukaadial acetate and ursolic acid acetate were sketched with the build panel in Maestro 12.0 [25]. The ligands were optimised using the ligprep component of Schrödinger Release 2019-2. The ligprep module generated a number of low energy structures with several ring conformations, stereochemistries, tautomers and ionization states that removed molecules with types of functional groups and specified numbers or molecular weights present with correct chiralities. The partial charges and ionization at biological pH 7.4 were obtained using the OPLS3e force field [24]. The obtained ligand structures were subjected to energy minimisation, which generated a low energy conformer of each ligand until their mean RMSD reached 0.001 Å.

### 2.7. Molecular Docking Studies

Iso-mukaadial acetate and ursolic acid acetate ligands were docked into the binding sites of *Plasmodium falciparum* and human hypoxanthine-guanine phosphoribosyl transferases protein using the glide docking module [26,27] as incorporated in the Schrödinger Release 2019-2 package. To relax the potential for the non-polar part of *Plasmodium falciparum* and human hypoxanthine-guanine phosphoribosyl transferases, the partial atomic charge and van der Waal radius were scaled by 1.00 and 0.25, respectively. The bound ligand in the 2VFA receptor was picked as a reference molecule to define the active pocket with 15 and 20 Å radius around the iso-mukaadial acetate and ursolic acid acetate ligands, respectively, for molecular docking. A receptor grid of X, Y and Z covering the iso-mukaadial acetate ligand was 22.919, 22.919 and 22.919 Å, while the grid dimensions of the centre grid box were 37.603, 37.603 and 37.603 Å for ursolic acid acetate molecule docking. The optimised compounds were docked flexibly into the 2VFA protein using the standard precision (SP) scoring function to evaluate the ligand–protein binding affinity. The output file generated in the form of the docking pose was visualised and analysed using the Maestro 11.8 Pose Viewer module.

### 2.8. Absorption, Distribution, Metabolism and Excretion/Toxicity (ADME/T) Prediction

A computational study of iso-mukaadial acetate and ursolic acid acetate compounds with 2VFA protein was carried out for the prediction of ADME/T parameters using the QikProp v6.0 in Maestro 12, Schrödinger 2019-2 [28]. The neutralisation of the molecules is essential before performing QikProp since it is not able to neutralise the compounds and generate descriptors.

### 2.9. Binding Free Energy Calculations

The local optimisation feature in Prime [22] was used to minimise the docked poses and the binding free energy (ΔG_bind_) of the complexes was simulated via the Prime/MM-GBSA module, which integrates the VSGB solvent model [29] and OPLS3e force field [24]. The ΔG_bind_ was estimated as follows [30,31]:Δ*G*_bind_ = Δ*G*_SA_ + Δ*G*_solv_ + Δ*E*(1)
where Δ*G*_SA_ is the difference between the surface area energy of the ligand–receptor complex and summation of the surface area energies for the isolated ligand and receptor. Δ*G*_solv_ is the difference between the Δ*G*_SA_ solvation energy of the ligand–receptor complex and summation of the solvation energies for the isolated ligand and receptor. Δ*E* is the difference between the minimised energies in the ligand–receptor complex. 

### 2.10. Molecular Dynamics Simulation 

The MD simulation was carried out on the minimised 2VFA complexes using the Desmond module [32,33] of Schrodinger Release 2019-2 with the OPLS3e force field. A simple point charge (SPC) water box [34,35], comprising of 14,024 and 11,743 water molecules for iso-mukaadial acetate and ursolic acid acetate complexes, respectively, was used for the solvation, and counter ions were added to neutralise the charge. A orthorhombic water box with a 10 Å buffer region between the box sides and the protein atoms was used to set up the system [36]. The complexes were minimised using the Broyden–Fletcher–Goldfarb–Shanno scheme with 41,667 iterations and a convergence threshold of 2.0 kcal/mol. A trajectory of 100 ns molecular dynamics simulation in the isothermal–isobaric (NPT) ensemble (barostat and thermostat relaxation time = 2.0 and 1.0 ps, respectively, T = 300 K, P = 1.01325 atm) was carried out using a Martyna–Tobias–Klein barostat [37] and a Nose–Hoover thermostat [38] to maintain constant pressure and temperature, respectively. An integration step of 2.0 and a cut-off radius of 9.0 Å was used to calculate the Coulombic interaction. The Ewald method [39] was employed to account for the long-range electrostatic interaction. The Simulation Interaction Diagram tool within Desmond was used to analyse the resulting trajectories.

## 3. Results and Discussion

### 3.1. Successful Expression and Purification Studies of Recombinant PfHGXPRT 

The expression of *Pf*HGXPRT using the pET-28a vector plasmid enables a tightly controlled gene expression system. High protein expression levels of *Pf*HGXPRT were achieved using previously described optimized conditions [40,41]. Upon inducing with IPTG, an increasing bandwidth was observed in the crude expression culture extract at the expected size of 28 kDa, and the gels were visualized using coomassie brilliant blue (Figure 1A: Lane 2HR to Lane O/N). The crude cell extracts were analysed by SDS-PAGE and confirmed by immunoblot analysis (Figure 1A); as expected, a protein band was evident at approximately 28 kDa (Figure 1A). The protein authenticity was confirmed by western blot using anti-polyhistidine horseradish peroxidase-conjugated antibodies probed with TMB as the substrate (Figure 1: Lower panel). The protein purification was performed under native conditions using affinity chromatography. The purified *Pf*HGXPRT protein showed a distinct band on 12.5% SDS-PAGE (Figure 1B). The average purified protein yield was estimated at 1.464 mg/mL using the NanoDrop® ND-1000 (Thermo Fisher Scientific, Waltham, MA, USA).

### 3.2. Iso-Mukaadial Acetate and Ursolic Acid Acetate Binds to PfHGXPRT

The protein–ligand and protein–protein interactions have an influence on enzyme complexes, biological signalling, DNA replication, transcription, translation and the overall homeostasis and maintenance of cellular entities [14]. Tyrosine residue exposure can be affected by the protein binding of the bioactive compounds, which can be seen as the absorbance of *Pf*HGXPRT shows high absorbance as all tyrosine residues are unoccupied. However, in the presence of the compound, fewer residues are exposed. In the presence of iso-mukaadial acetate, the highest protein concentration illustrates the lowest absorbance, while in the absence of the absorbance, peak shifts were clearly observed in the results, thus providing evidence that iso-mukaadial acetate and ursolic acid acetate directly interact with *Pf*HGXPRT in a dose-dependent manner. Furthermore, the results imply that the interaction mechanism of action may be similar to the commercial inhibitor, thus inducing a comparative structural conformational change and permitting the interaction with aromatic side chains (Figure 2). 

### 3.3. MST Binding Analysis of Ursolic Acid Acetate and Iso-Mukaadial Acetate against PfHGXPRT 

A number of methods have been developed to enable the study of protein interactions which include native mass spectrometry, fluorescence spectroscopy, static or dynamic light scattering, biolayer interferometry [14,42,43]. However, all these methods have limitations such as immobilization, buffer restrictions, mass transport, small peptide interactions and most crucial being the sample consumption [14,40]. Unlike other methods for studying the interactions of proteins, MST is quality controlled, has no need for sample immobilization, has a low sample consumption and a free choice of buffers, and is highly accurate and fast [14,15]. In this study, the interaction between the ligands and the *Pf*HGXPRT was further evaluated using MST. The fluorescently labelled *Pf*HGXPRT protein had an altered steady-state and was mobility dependent upon the ligand used during binding tests. The binding affinity was expressed as KD, and the data analysis was carried out using MO. Affinity Analysis software (NanoTemper Technologies, Munich, Germany) version 2.1.3. MST operates using a temperature gradient which molecules move along which causes alterations in charge, size and hydration shell which can be measured as they are unique to each protein and its interaction with macromolecules, which affects their movement along the temperature gradient [14]. In this study, strong binding properties were observed between *Pf*HGXPRT and the ligands, this is characterised by the good splitting observed; the lowest KD (1.5272 × 10^−12^) and highest KD confidence (±2.193 × 10^−9^). The increased binding capabilities *Pf*HGXPRT can be attributed to the formation of new cis- and trans-conformational change resulting in better interaction and peptide bonds formed in the binding site between the ligand and the receptor [7]. The observed data indicate that conformational changes in the *Pf*HGXPRT hydration shell, charge and surface area which alter mobility are induced by the MA and UAA. The application of this MST method to *Pf*HGXPRT allowed us, for the first time, to determine the affinity of the complex formed between *Pf*HGXPRT and the ligands; IMA and UAA (Table 1).

*Pf*HGXPRT showed the strongest binding properties towards the iso-mukaadial acetate compound indicated by the lowest *K_D_* (0.0833 µM) and the highest *K_A_* constant (12.0058). Whilst *Pf*HGXPRT showed the lowest binding properties towards the ursolic acid acetate compound indicated by the lowest *K_D_* (2.8396 µM) and the highest *K_A_* constant (0.3522 M^−1^).

### 3.4. Molecular Docking Study 

The in vitro studies carried out using iso-mukaadial acetate and ursolic acid acetate molecules with *Plasmodium falciparum* hypoxanthine-guanine phosphoribosyl transferases protein were promising for antimalarial activity. Thus, binding interaction studies between the compounds and *Pf*HGXPRT were further supported via an in silico molecular docking simulation. The interactions of iso-mukaadial acetate and ursolic acid acetate compounds with the 2VFA protein were considered to evaluate the binding affinity, energy and activity of the considered compounds. The docking study confirmed that the iso-mukaadial acetate and ursolic acid acetate compounds were fully docked strongly to the receptor site (Figure 3).

The ocking results between the screened ligands and the 2VFA receptor are summarised in Table 2.

The docking analysis of iso-mukaadial acetate and ursolic acid acetate with 2VFA showed docking scores of −4.10 and −2.96 kcal/mol, respectively. The human HGPRT had a docking score of −3.28 and −2.89 kcal/mol, respectively, for iso-mukaadial acetate and ursolic acid acetate compounds. The oxygen atom of the iso-mukaadial acetate and ursolic acid acetate compounds formed a hydrogen bond with VAL187 with the bond length 1.98 and 2.06 Å, respectively, at the pocket sites of human HGPRT. The docking of iso-mukaadial acetate in the binding pockets revealed that iso-mukaadial acetate interacts more effectively than ursolic acid acetate. Therefore, the antimalarial activity was predicted to be better in iso-mukaadial acetate than ursolic acid acetate ligand. The docking of the synthesised ligands with the 2VFA receptor shows bonds with several amino acid residues, which are within the active sites of the receptor. The binding pocket was well thought out to be the active site where iso-mukaadial acetate and ursolic acid acetate complexes with the 2VFA receptor. The protein–ligand interaction patterns of both complexes suggest that these amino acids are involved in the formation of several hydrogen bonds between the ligands. Hydrogen bonding interactions serve as anchors to guide the orientation of the ligands in its binding pocket, thereby supporting the electrostatic and steric interaction. Moreover, the interactions between the 2VFA receptor and iso-mukaadial acetate or ursolic acid acetate are because of non-covalent interactions, as well as hydrogen bonds to achieve suitable binding. Iso-mukaadial acetate forms hydrogen bonds with the 2VFA receptor and the interacting amino acid residues of ASN47, LYS51, ARG80, GLY81, THR84, ALA85, TYR201, SER202, ARG210, ASP211 and ASP213, which is shown in Figure 3a.

The amino acid residues of the 2VFA protein that interacted with ursolic acid acetate are GLU69, HIE71, LYS103, PRO104, PHE106, GLY107, HIE109, GLU130, ASP131, CYS134, LEU135 and LYS138. Hydrophobic and hydrogen bonding interactions between iso-mukaadial acetate and amino acid residues of the 2VFA receptor play a significant contribution in the stabilisation of the ligand conformation at the active sites. In this hydrophobic interaction, ALA85 and TYR201 were making a hydrophobic interaction with the iso-mukaadial acetate. Binding analysis of iso-mukaadial acetate depicted a hydrogen bond interaction between ARG80 and ASP213 residues at a distance of 1.66 and 1.52 Å, respectively. On the other hand, ursolic acid acetate forms hydrogen bonds with the LYS138 residue at a bond length of 1.92 Å (Figure 4b). Ursolic acid acetate also forms a salt bridge with the LYS138 residue at a bond length of 4.87 Å. Moreover, PRO104, PHE106, CYS134 and LEU135 residues are in the hydrophobic contact. Moreover, there is less hydrogen bonding in ursolic acid acetate complex than the iso-mukaadial acetate complex.

### 3.5. Binding Free Energy Analysis

The binding affinity of the ligands was predicted by a theoretical study that combined binding free energy calculation, molecular docking and dynamics. The binding energy of iso-mukaadial acetate-2VFA and ursolic acid acetate-2VFA complexes was predicted as -44.46 and -26.73 kcal/mol, respectively, see Table 2. Moreover, the 2VFA complexes were found to have a stronger binding affinity than the 1Z7G complexes (-7.16 and -13.70 kcal/mol for iso-mukaadial acetate-1Z7G and ursolic acid acetate-1Z7G complexes, respectively), suggesting that iso-mukaadial acetate and ursolic acid acetate compounds appeared to be a better inhibitor of the 2VFA than the 1Z7G receptor. The observed negative binding free energies established the good binding affinity and stability of both complexes. The better inhibitor of iso-mukaadial acetate relative to the ursolic acid acetate ligand is evident from the pIC_50_ values (iso-mukaadial acetate has a higher pIC_50_ value relative to ursolic acid acetate ligand). The strong thermodynamic interactions of iso-mukaadial acetate and ursolic acid acetate with 2VFA contributes to their good in silico binding affinity and offers in-depth insights into observed in vitro antimalarial activity.

### 3.6. Predicted ADME/T Properties

The pharmaceutically relevant properties and physiochemical descriptors of the identified lead molecules against 2VFA are summarised in Table 3. 

As shown in Table 3, the most potent iso-mukaadial acetate ligand displays a favourable rule of five parameters equal to 0, while the ursolic acid acetate ligand exhibits rule of five parameters equal to 1. Therefore, most of the physicochemical descriptors are within the tolerable value accepted for human use. Therefore, the Qikprop module predicted that the iso-mukaadial acetate ligand is exempt from carcinogenesis and skin irritations, as well as having an acceptable QP log *P*_o_/*w* value. Thus, we concluded that the iso-mukaadial acetate compound had good drug-likeness candidates and passed Lipinski’s rule of five and Jorgensen’s rule of three.

### 3.7. MD Simulation 

To monitor the structural changes and stability of the 3D model of iso-mukaadial acetate and ursolic acid acetate complexes, MD simulation was carried out. The protein backbone frames were aligned to the backbone of the original structure and then the RMSD deviation was calculated with respect to the original structures, as given in Figure 5.

The RMSD shows that the conformations of these two complexes do not reach equilibrium until about 30 ns (Figure 5a), and the mean backbone C_α_ RMSD values for iso-mukaadial acetate and ursolic acid acetate complexes were 1.89 ± 0.15 Å and 2.21 ± 0.24 Å, respectively. The above result suggests that the fluctuation of the iso-mukaadial acetate complex is less significant than that of the ursolic acid acetate complex. The drift in the initial 30 ns was because of the absence of restraints during the MD simulation process. Moreover, after the simulation time of 30 ns, both complexes exhibited a steady trajectory of up to 100 ns. The RMSD ranging from 1.75 to 2.10 Å and 2.28 to 2.90 Å for iso-mukaadial acetate and ursolic acid acetate complexes, respectively, signified the stability of the complexes during the 100 ns simulations. The average RMSD for iso-mukaadial acetate complex was 1.98 Å with an SD of 0.07 Å, while the ursolic acid acetate complex was 2.60 Å with an SD of 0.11 Å. Thus, the binding pocket of iso-mukaadial acetate exhibits less fluctuation than that of ursolic acid acetate, which indicates that iso-mukaadial acetate may form stronger interactions with the 2VFA receptor compared to ursolic acid acetate. This phenomenon is in accordance with experimental bioactivities.

Moreover, the root-mean-square fluctuation (RMSF) against the residue number of the protein–ligand complex was used to examine the flexibility and stability of individual residues (Figure 6). 

We found that most of the amino acid residues were fluctuated by less than 1.5 Å, which was consisted with other studies. As shown in Figure 6, the two complexes share similar trends of dynamic features and similar RMSF distributions. The RMSF of only a few residues located in the non-active site exceeded 2.0 Å, suggesting that most of the residues were stable. In this study, a lower degree of conformational changes in the side chains was observed, since the RMSF of the amino acid residues was below 4.0 Å during the entire simulation (Figure 6a). Much reduction in the fluctuation was observed when the 2VFA receptor was bound to iso-mukaadial acetate. We clearly observed relatively smaller dynamic fluctuations of ASN47, LYS51, ARG80, GLY81, THR84, ALA85, TYR201, SER202, ARG210, ASP211 and ASP213 residues located in the active site of iso-mukaadial acetate-2VFA complex. However, the regions around the ASP98, THR119, ASP128, ASN180 and ASN204 residues exhibited much larger dynamic fluctuations since they were located in the non-active site. The region of relatively small fluctuations (GLU69, HIE71, LYS103, PRO104, PHE106, GLY107, HIE109, GLU130, ASP131, CYS134, LEU135 and LYS138) can be assigned to the active sites of the ursolic acid acetate-2VFA complex. We observed that the regions around the ASP98, GLN117, GLY127, GLN160, GLY181 and TYR203 residues exhibit much larger dynamic fluctuations since they are located in the non-active sites. The overall fluctuation of iso-mukaadial acetate complex was lower than the ursolic acid acetate complex. This may be because of the smaller size of iso-mukaadial acetate, which allowed it to move freely in the active sites. Therefore, a tightly interacting compound (e.g., ursolic acid acetate) may perhaps change the overall flexibility and stability of the 2VFA receptor, resulting in increased RMSF values. The RMSF plot of iso-mukaadial acetate and ursolic acid acetate compounds in Figure 6b show the highest flexibility of up to above 4 Å, which was due to their flexibility within the pocket sites of 1Z7G receptor. 

Various inter-molecular interactions, such as electrostatic, ionic, hydrogen bond and hydrophobic interactions, were formed between the ligand and receptor during the MD simulation. This makes the ligand well stabilised in the active sites. The interaction of hits with iso-mukaadial acetate-2VFA and ursolic acid acetate-2VFA complexes higher than 30% after theMD simulation is shown in Figure 7.

The MD simulation also confirmed the location of the binding modes of both complexes. All the protein–ligand interactions observed in the docking study were retained throughout the entire MD simulation, indicating stable binding of 2VFA with iso-mukaadial acetate and ursolic acid acetate ligands. The observed stability may be responsible for the high in vitro antimalarial potency of the iso-mukaadial acetate-2VFA and ursolic acid acetate-2VFA complexes. The MD study revealed that iso-mukaadial acetate forms hydrogen bonds with THR84 and ARG80 via a water molecule and ASP213 residue with direct hydrogen bonding. Ursolic acid acetate interacted with GLU69 via a water molecule, while a direct hydrogen bonding with LYS103 and LYS138 was observed. In addition to water molecule with LYS165, hydrophobic interactions with MET94, VAL187 and TYR194 residues were the major contributors towards the binding of iso-mukaadial acetate and ursolic acid acetate ligands with the 1Z7G receptor (Figure 7c,d).

## 4. Conclusions 

In summary, the study suggests further study of the *Pf*HGXPRT protein as a key enzyme in the malaria parasite, as it depends upon it for nucleotide synthesis. The inhibitory activity of iso-mukaadial acetate and ursolic acid acetate triggers an enhanced interest in the use of traditional medicinal plant-derived drugs as potential malarial chemotherapeutic candidates. The trend in the observed antimalarial activity was further rationalised by molecular docking studies to examine whether the active iso-mukaadial acetate and ursolic acid acetate compounds could target the 2VFA receptor. The molecular docking studies supported the in vitro antimalarial results and the stability of iso-mukaadial acetate and ursolic acid acetate complexes, confirmed by MD studies, benefits the experimental observation, thereby predicting that this class of compounds may serve as inhibitors for the 2VFA receptor. The Prime/MMGBSA binding free energy results suggest stronger binding affinity in the selected screened ligands towards the 2VFA receptor and the in silico ADME/T properties of the synthesised compounds revealed their drug-like properties. The favourable molecular characteristics based on the Lipinski rule of five, the high binding free energies and the docking score accounted for the high antimalarial activity of the iso-mukaadial acetate complex.

## Figures and Tables

**Figure 1 biomolecules-09-00861-f001:**
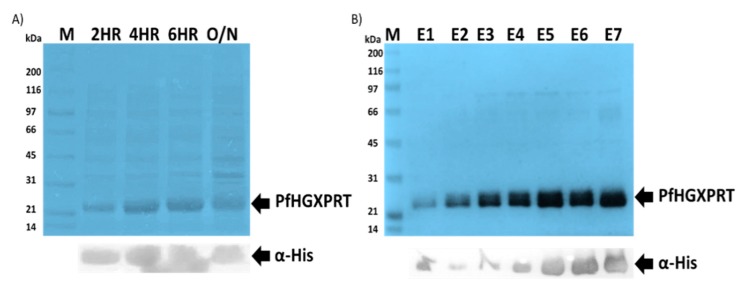
Recombinant Plasmodium falciparum hypoxanthine-guanine-xanthine phosphoribosyltransferase (*Pf*HGXPRT) expression and purification. Isopropyl-β-D-1-thiogalactopyranoside (IPTG)-induced protein expression (**A**) and affinity purification (**B**) of *Pf*HGXPRT was visualized by sodium dodecyl sulphate–polyacrylamide gel electrophoresis (SDS-PAGE) 12.5% (Upper panel) and confirmed by western blot (lower panel) using α-His horse-raddish-peroxidase (HRP)-conjugated antibodies (Sigma-Aldrich, St. Louis, MO, USA). Lane M represents a broad range molecular marker (Bio-Rad) in kilodaltons (kDa) shown on the left-hand side of the SDS-PAGE images. Lane: 2HR, 4HR, 6HR and O/N, respectively, represent cell lysates collected at 2, 4, 6 and 24 h post-IPTG induction. Lane E1–E7 represents the successfully purified *Pf*HGXPRT protein elutions.

**Figure 2 biomolecules-09-00861-f002:**
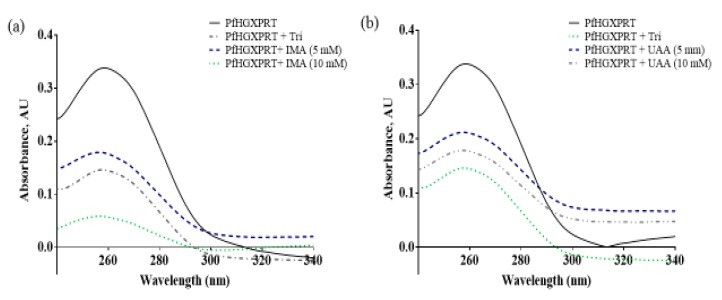
UV-Vis absorption spectra of *Pf*HGXPRT in the presence of triethyl-2-phosphonobutyrate (Tri) standard inhibitor with (**a**) iso-mukaadial acetate (IMA) and (**b**) ursolic acid acetate (UAA).

**Figure 3 biomolecules-09-00861-f003:**
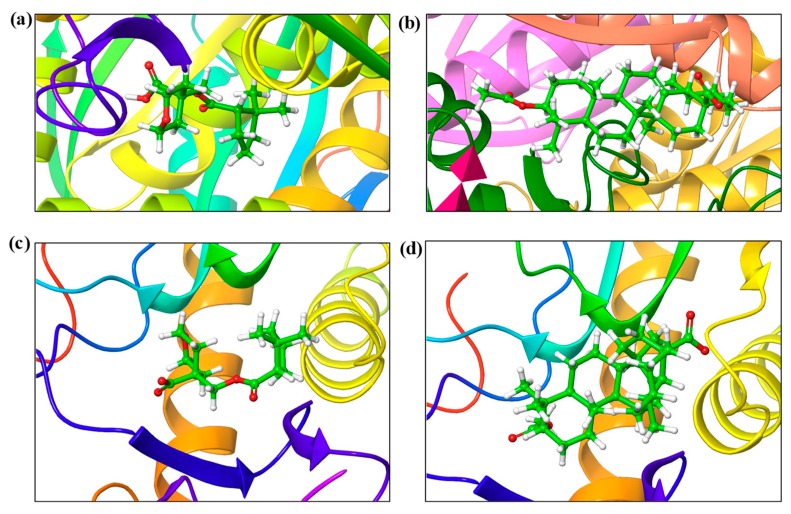
The binding modes of (**a**) iso-mukaadial acetate and (**b**) ursolic acid acetate ligands with the 2VFA receptor. (**c**) iso-mukaadial acetate and (**d**) ursolic acid acetate ligands with 1Z7G receptor.

**Figure 4 biomolecules-09-00861-f004:**
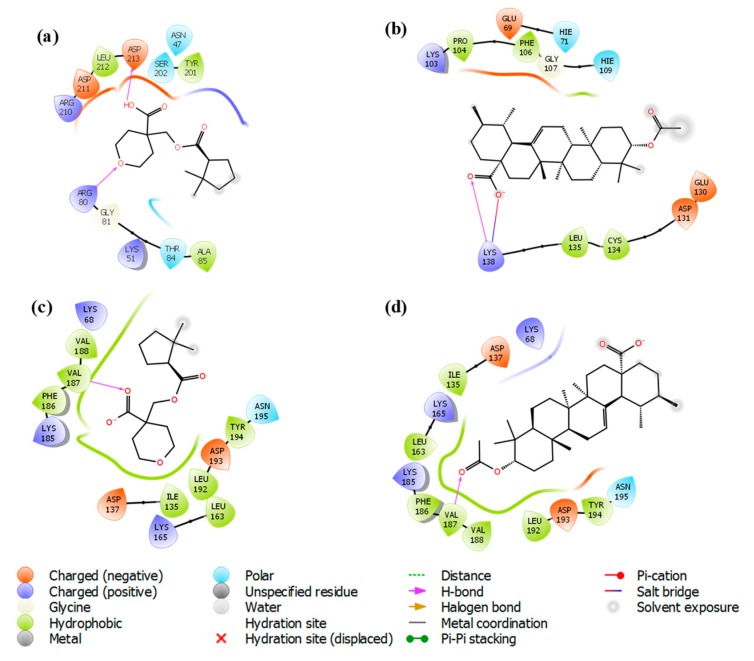
Two-dimensional protein-ligand interaction of (**a**) iso-mukaadial acetate and (**b**) ursolic acid acetate compounds with hydrogen bond interactions at the pocket sites of the 2VFA receptor. (**c**) iso-mukaadial acetate and (**d**) ursolic acid acetate compounds with hydrogen bond interactions at the pocket sites of the 1Z7G receptor.

**Figure 5 biomolecules-09-00861-f005:**
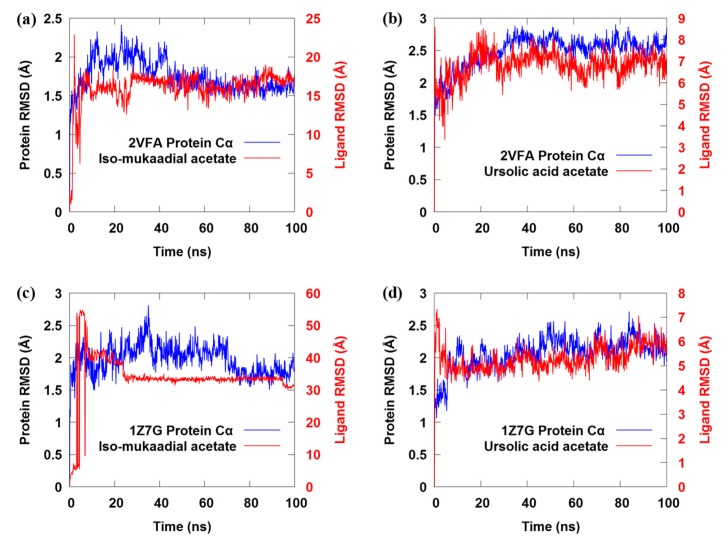
(**a**–**d**) The backbone C_α_ root-mean-square deviation (RMSD) profile for both iso-mukaadial acetate and ursolic acid acetate complexes over 100 ns MD trajectories.

**Figure 6 biomolecules-09-00861-f006:**
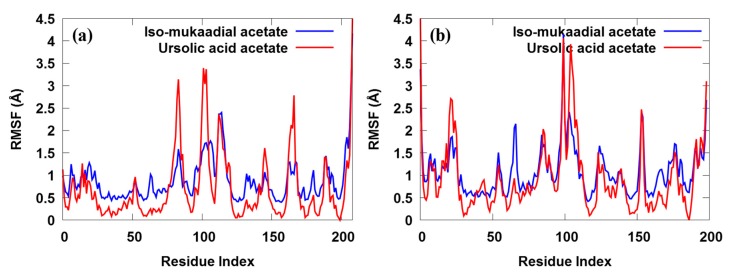
The ligand root-mean-square fluctuation for both iso-mukaadial acetate and ursolic acid acetate towards the (**a**) 2VFA and (**b**) 1Z7G receptors over 100 ns MD trajectories.

**Figure 7 biomolecules-09-00861-f007:**
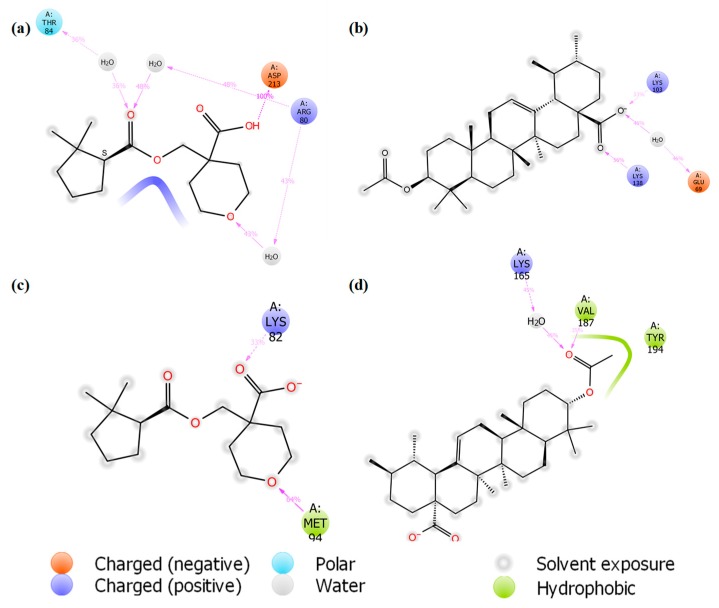
Interactions of (**a**) iso-mukaadial acetate and (**b**) ursolic acid acetate with the binding pocket residues of the 2VFA receptor after 100 ns MD simulation. (**c**) Iso-mukaadial acetate and (**d**) ursolic acid acetate interactions with the binding site residues of the 1Z7G receptor during the 100 ns MD simulation timeframe.

**Table 1 biomolecules-09-00861-t001:** *Pf*HGXPRT binding affinities and binding constants of bioactive compounds iso-mukaadial acetate (IMA) and ursolic acid acetate (UAA).

Protein and Ligand	*K_D_* (µM)	*K_A_* (M^−1^)	*Binding Energy (Kcal/mol)*
*Pf*HGXPRT + IMA	0.0833	12.0058	−11.66
*Pf*HGXPRT + UAA	2.8396	0.3522	−13.78

*K_A_* = 1/*K_D_**_._*

**Table 2 biomolecules-09-00861-t002:** Docking scores and interaction residues for the binding mode of iso-mukaadial acetate and ursolic acid acetate ligands with the 2VFA receptor.

Complex	Docking Scores	pIC50	Δ*G*_bind_	Residue Involved H-bond
Iso-mukaadial acetate-2VFA	−4.10	5.35	−26.73	ARG80, ASP213
Ursolic acid acetate-2VFA	−2.96	4.19	−44.46	LYS138

**Table 3 biomolecules-09-00861-t003:** The principle descriptors and physicochemical properties of identified IMA and UAA compounds towards the 2VFA receptor.

Principal Descriptors	IMA	UAA	(Range 95% of Drugs)
Solute Molecular Weight	284.352	498.745	(130.0–725.0)
Solute Dipole Moment	1.465	7.944	(1.0–12.5)
Solute Total SASA	518.477	741.364	(300.0–1000.0)
Solute Hydrophobic	408.528	628.533	(0.0–750.0)
Solute Hydrophilic	109.949	104.637	(7.0–330.0)
Solute Carbon Pi SASA	0	8.195	(0.0–450.0)
Solute Weakly Polar SASA	0	0	(0.0–175.0)
Solute Molecular Volume (Å^3^)	934.962	1522.361	(500.0–2000.0)
Solute vdW Polar SA	81.308	75.126	(7.0–200.0)
Solute Number of Rotatable Bonds	4	2	(0.0–15.0)
Solute as Donor - Hydrogen Bonds	1	1	(0.0–6.0)
Solute as Acceptor - Hydrogen Bonds	5.7	4	(2.0–20.0)
Solute Globularity (Sphere = 1)	0.892	0.863	(0.75–0.95)
Solute Ionization Potential (eV)	10.803 *	9.669	(7.9–10.5)
Solute Electron Affinity (eV)	−0.815	−0.737	(−0.9–1.7)
Predictions for Properties			
QP Polarizability (Å^3^)	28.695M	53.565M	(13.0–70.0)
QP log P forhexadecane/gas	8.065M	13.173M	(4.0–18.0)
QP log P foroctanol/gas	13.452M	21.994M	(8.0–35.0)
QP log P forwater/gas	7.911M	7.234M	(4.0–45.0)
QP log P foroctano/water	2.561	7.020 *	(−2.0–6.5)
QP log S foraqueous solubility	−3.104	−7.984 *	(−6.5–0.5)
QP log S - conformation independent	−2.765	−7.823	(−6.5–0.5)
QP log K hsa Serum Protein Binding	−0.231	1.771 *	(−1.5–1.5)
QP log BB for brain/blood	−0.599	−0.493	(−3.0–1.2)
No. of Primary Metabolites	2	2	(1.0–8.0)
HERG K+ Channel Blockage: log IC_50_	−1.474	−1.976	(concern below −5)
Apparent Caco-2 Permeability (nm/sec)	227	255	(<25 poor, >500 great)
Apparent MDCK Permeability (nm/sec)	126	143M	(<25 poor, >500 great)
QP log Kp for skin permeability	−3.162	−3.227	(Kp in cm/hr)
Jm, max transdermal transport rate	0.154	0	(micrograms/cm^2^-hr)
Lipinski Rule of 5 Violations	0	1	(maximum is 4)
Jorgensen Rule of 3 Violations	0	1	(maximum is 3)
% Human Oral Absorption in GI (±20%)	84	100	(<25% is poor)
Qualitative Model for Human Oral Absorption	High	Low	(>80% is high)

* indicates a violation of the 95% range; M indicates MW is outside training range.

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
