# Peer review of "Evaluating Iso-Mukaadial Acetate and Ursolic Acid Acetate as Plasmodium falciparum Hypoxanthine-Guanine-Xanthine Phosphoribosyltransferase Inhibitors"

_biomolecules, 2019, doi:10.3390/biom9120861_

Round 1

Reviewer 1 Report

To elucidate the unknown mode of action of natural antimalarials - iso-mukaadial acetate and ursolic acid - this work describes several in silico and in vitro approaches to prove an involvement of PfHGXPRT, the key enzyme of the purine salvage pathway, in activity of these compounds. This could be of general interest.

Experiments and methods are sufficiently described, the focus of the manuscript is mainly in computational work.

- Binding activity of the compounds was evaluated in enzyme experiments with recombinant PfHGXPRT and human HGPRT. Binding properties to human counterpart are just briefly mentioned on the page 8. It could be better to include them to Table 1 for direct comparison with the PfHGXPRT data.

- The authors proved, that the binding of the both compounds is dose dependent and cause similar conformational changes of the enzyme as the commercial inhibitor (triethyl 2-phosphonobutyrate). But there is no direct proof, that the compounds inhibit reaction catalysed by the phosphoribosyltransferase. Such experiment would significantly increase the value of the study and would strongly support the suggested mode of action. Authors should comment on this in the manuscript.

- To further support the hypothesis of mode of action via inhibition of PfHGXPRT, extensive in silico study is reported. For the docking study, crystal structure of a chimera of PfHGXPRT and human HGPRT (PDB code 2VFA, resolution 2.8 A) is used. The character of the nature of the chimeric enzyme (core of the protein from the human enzyme and the hood region from the Pf enzyme) should be discussed when analysing in silico data.

Authors should explain in the manuscript, why they use this chimera instead of the known crystals of real PfHGXPRT (PDB codes 3OZF or 3OZG, Hazleton, et al. 2012 Chem Biol 19 721-730) that are more relevant and have much better resolution.

-The comparison of the in silico binding of the studied compounds between real PfHGXPRT and human HGPRT could be also included to show if there could be any selectivity over the human counterpart during the treatment. There is plenty of suitable crystals of human HGPRT published, some of them even in complex with antiplasmodial inhibitors (e.g. Spacek et al. J.Med.Chem. 2017, 7539-7554 or Keough et al. J.Med.Chem. 2015).

-Page 8:  the sentence “Thus the antimalarial activity of plasmodium falciparum and human hypoxanthine-guanine phosphoribosyl transferases was further supported via in silico molecular docking simulation.” does not make any sense.

Author Response

Comment: Binding activity of the compounds was evaluated in enzyme experiments with recombinant PfHGXPRT and human HGPRT. Binding properties to human counterpart are just briefly mentioned on the page 8. It could be better to include them to Table 1 for direct comparison with the PfHGXPRT data.

Response: No Human HGPRT interaction data was reported in this paper, this was a typing error in this instance. This has been corrected. 

Comment: The authors proved, that the binding of the both compounds is dose dependent and cause similar conformational changes of the enzyme as the commercial inhibitor (triethyl 2-phosphonobutyrate). But there is no direct proof, that the compounds inhibit reaction catalysed by the phosphoribosyltransferase. Such experiment would significantly increase the value of the study and would strongly support the suggested mode of action. Authors should comment on this in the manuscript.

Response: We would like to thank the reviewer for this point, however, the current manuscript only sought to investigate the interaction and binding affinity between the purified proteins and the compounds, as part of phase 2 of the project we will investigate direct inhibition using MST,  ITC and deuterium exchange mass spec using PfHGXPRT subdomains to investigate the exact residues involved in the protein-compound interaction and activity inhibition.  

Comment: To further support the hypothesis of mode of action via inhibition of PfHGXPRT, extensive in silico study is reported. For the docking study, crystal structure of a chimera of PfHGXPRT and human HGPRT (PDB code 2VFA, resolution 2.8 A) is used. The character of the nature of the chimeric enzyme (core of the protein from the human enzyme and the hood region from the Pf enzyme) should be discussed when analysing in silico data.

Response: Such suggestion has been considered in the current manuscript. The crystal structure of a chimera of PfHGXPT, which consists of the core of the hood region from the Pf enzyme and the protein from the human enzyme, has been determined as a complex with the product guanosine monophosphate [16]. The chimera can utilise xanthine, guanine and hypoxanthine, as substrates, which is similar to the Plasmodium falciparum enzyme. The chimera occurs as a monomer-dimer mixture in solution, but shifts to a tetramer with the addition of phosphoribosyl pyrophosphate [16].

Comment: Authors should explain in the manuscript, why they use this chimera instead of the known crystals of real PfHGXPRT (PDB codes 3OZF or 3OZG, Hazleton, et al. 2012 Chem Biol 19 721-730) that are more releva nt and have much better resolution.

Response: Solution studies on 2VFA have shown that it occurs as a mixture of monomer and dimer, and undergoes tetramerisation on the addition of phosphoribosyl pyrophosphate [16]. On the other hand, both PfHGXPT and human are tetramers in all liganded states. 2VFA also shows additional specificity for xanthine, similar to Domain Switch 1 (DS1) and PfHGPRT. DS1 was the first studied chimeric enzyme, which contained the 49 terminal amino acids residues from PfHGXPT and the rest from human HGPRT [19].

Comment: The comparison of the in silico binding of the studied compounds between real PfHGXPRT and human HGPRT could be also included to show if there could be any selectivity over the human counterpart during the treatment. There is plenty of suitable crystals of human HGPRT published, some of them even in complex with antiplasmodial inhibitors (e.g. Spacek et al. J.Med.Chem. 2017, 7539-7554 or Keough et al. J.Med.Chem. 2015).

Response: For comparison, molecular docking and molecular dynamics studies on human HGPRT was considered. The crystal structure of the free Human Hypoxanthine-guanine Phosphoribosyltransferase (HGPRT) receptor with PDB-ID: 1Z7G [18]) was obtained from the RCSB Protein Data Bank (https://www.rcsb.org/structure) [17]. We have provided data on the molecular docking and molecular dynamics and they are highlighted in the manuscript.

Comment: Page 8:  the sentence “Thus the antimalarial activity of plasmodium falciparum and human hypoxanthine-guanine phosphoribosyl transferases was further supported via in silico molecular docking simulation.” does not make any sense.

Response: We completely agree with the reviewer, since antimalarial studies were not conducted in this report. Thus, the sentence was removed accordingly.

Reviewer 2 Report

The manuscript entitled “Evaluating Iso-Mukaadial Acetate and Ursolic Acid Acetate as Plasmodium falciparum  Hypoxanthine-Guanine-Xanthine  Phosphoribosyltransferase Inhibitors” describes the selective inhibitory and binding action of iso-mukaadial acetate and ursolic acid acetate against recombinant PfHGXPT  using in-silico and experimental approaches. In addition, the mode of action for the observed antimalarial activity was further established by molecular docking study. The author claimed that this finding provide a basis for the recommendation of iso-mukaadial acetate and ursolic acid acetate as high-affinity ligands and drug candidates against PfHGXPT. However, in my opinion, the novelty of these experiments are not sufficient enough to be published in Biomolecules. Therefore, this reviewer cannot recommend this manuscript for publication in Biomolecules.

Author Response

Comment: “… in my opinion, the novelty of these experiments are not sufficient enough to be published in Biomolecules”.

Response: The authors would like to appreciate the time taken by the reviewer to read the manuscript however, no specific concerns were raised.

Reviewer 3 Report

This work stems from initial findings that iso-mukaadial acetate and ursolic acid acetate, are relatively active antimalarials inhibiting PfHGXPT (hypoxanthine-guanine-xanthine phosphoribosyltransferase) in a dose-dependent manner.

Here, the authors entered to investigate the selective inhibitory and binding action of these two compoundsagainst recombinant PfHGXPT using in-silico (MD, ADME, etc) and experimental approaches (KD, strong binding).

Interesting in silico results are good basis for further studies in this area: toxicity, cure effect, PK, etc.

Author Response

No negative issues were raised by this reviewer and it appears the reviewer also recommends the publication of the manuscript. We sincerely appreciate the time the reviewers took to read the manuscript.

Round 2

Reviewer 2 Report

Considering the concerns of all reviewers and author's responses, the manuscript can be accepted for publication in Biomolecules.